# Single-Domain Antibodies Represent Novel Alternatives to Monoclonal Antibodies as Targeting Agents against the Human Papillomavirus 16 E6 Protein

**DOI:** 10.3390/ijms20092088

**Published:** 2019-04-28

**Authors:** Melissa Togtema, Greg Hussack, Guillem Dayer, Megan R. Teghtmeyer, Shalini Raphael, Jamshid Tanha, Ingeborg Zehbe

**Affiliations:** 1Biotechnology Program, Lakehead University, Thunder Bay, ON P7B 5E1, Canada; mtogtema@lakeheadu.ca; 2Probe Development and Biomarker Exploration, Thunder Bay Regional Health Research Institute, Thunder Bay, ON P7B 6V4, Canada; gdayer@lakeheadu.ca (G.D.); mteghtmeyer2@gmail.com (M.R.T.); 3Human Health Therapeutics Research Centre, National Research Council Canada, Ottawa, ON K1A 0R6, Canada; greg.hussack@nrc-cnrc.gc.ca (G.H.); shalini.raphael@nrc-cnrc.gc.ca (S.R.); 4Department of Biology, Lakehead University, Thunder Bay, ON P7B 5E1, Canada; 5School of Environmental Sciences, University of Guelph, Guelph, ON N1G 2W1, Canada; 6Department of Biochemistry, Microbiology and Immunology, University of Ottawa, Ottawa, ON K1H 8M5, Canada

**Keywords:** Human papillomavirus, E6 oncoprotein, cancer, single-domain antibody, VHH, phage display, nanobody

## Abstract

Approximately one fifth of all malignancies worldwide are etiologically associated with a persistent viral or bacterial infection. Thus, there is a particular interest in therapeutic molecules which use components of a natural immune response to specifically inhibit oncogenic microbial proteins, as it is anticipated they will elicit fewer off-target effects than conventional treatments. This concept has been explored in the context of human papillomavirus 16 (HPV16)-related cancers, through the development of monoclonal antibodies and fragments thereof against the viral E6 oncoprotein. Challenges related to the biology of E6 as well as the functional properties of the antibodies themselves appear to have precluded their clinical translation. Here, we addressed these issues by exploring the utility of the variable domains of camelid heavy-chain-only antibodies (denoted as VHHs). Through construction and panning of two llama, immune VHH phage display libraries, a pool of potential VHHs was isolated. The interactions of these with recombinant E6 were further characterized using an enzyme-linked immunosorbent assay (ELISA), Western blotting under denaturing and native conditions, and surface plasmon resonance. Three VHHs were identified that bound recombinant E6 with nanomolar affinities. Our results lead the way for subsequent studies into the ability of these novel molecules to inhibit HPV16-infected cells in vitro and in vivo.

## 1. Introduction

Approximately 20% of cancers worldwide are etiologically associated with persistent infection by microbes such as viruses [1,2]. Advantageously, when prophylactic measures are unavailable or ineffective, the opportunity remains to specifically target infected cells using therapeutic molecules which block the expression or function of oncogenic microbial proteins. As these proteins often bear minimal homology to those of the host, such molecules are anticipated to have fewer off-target effects on surrounding uninfected cells compared to conventional treatments, reducing treatment-associated toxicity and facilitating disease intervention at the earliest stages of lesion development [3].

One of the most ubiquitous tumor viruses in humans is the papillomavirus (HPV). Due to its common transmission through sexual contact, almost all individuals will become infected with this keratinocyte-tropic DNA virus throughout their lifetime [4,5]. Twelve high-risk mucosal HPV types have been well-established as the causative agents in almost all cases of cervical cancer [6] and have been implicated in anogenital as well as oropharyngeal cancers [7]. Of these, HPV16 is the most commonly identified [7,8]. Most strikingly, the incidence of HPV-related oropharyngeal cancers in males has been on the rise in Canada, the United States, and Europe, outpacing that of HPV-related cervical cancers [9,10,11]. Of the products encoded by HPV16′s episomal genome (Figure 1A) [12], the E6 oncoprotein (~18 kDa) (Figure 1B) promotes the immortalization and malignant transformation of persistently infected cells through a variety of interactions with host intracellular proteins [13,14], including degradation of the tumor suppressor protein p53. Indeed, in HPV-infected keratinocytes, cancer is caused by the dual activity of the two oncoproteins E6 and E7. The latter stimulates uncontrolled cell proliferation which in turn activates the tumor suppressor protein p53. The infected cell is protected from apoptosis by the E6 protein which degrades p53 inactivating the pro-apoptotic signal [15]. Hence we hypothesize that molecules inactivating E6 could restore apoptosis. As E6 is expressed in both precancerous lesions and tumors, and its inhibition can promote apoptosis when expression of the pro-proliferative E7 oncoprotein is retained, it has accordingly been recognized as an optimal HPV16-specific therapeutic target [3].

Inhibitory molecules of a transient nature may present fewer ethical concerns than those intended to permanently edit the viral genome (e.g., CRISPR/Cas9 [19]). They can be designed to target either E6 mRNA, preventing its translation, or the E6 protein itself, sterically hindering its intracellular interactions. Molecules which mimic components of a natural immune response are of particular interest as they use endogenous cellular pathways and are hypothesized to have high functionality with low toxicity [20,21]. However, transcript silencing using synthetic small interfering ribonucleic acids (siRNAs) has been limited by challenges in achieving the desired therapeutic effects at clinically relevant concentrations, off-target effects, as well as siRNA stability and uptake in vivo [21]. Alternatively, monoclonal antibodies (mAbs) benefit from greater therapeutic specificity and longer intracellular half-lives than siRNA [22]. Several mAbs specific to the N-terminal region (clones 1F1, 6F4, 4C6) or the second zinc-binding domain (clones 1F5, 3B8, 3F8) of the HPV16 E6 protein have been isolated from immunized mice [23,24,25]. Preliminary studies by both us and others have demonstrated that when transiently transfected into HPV16-positive cell cultures, these mAbs elicited a notable restoration of p53 protein levels [20,26] and that conjugation of the mAbs to a nuclear localization signal (NLS) improved their ability to access E6′s mainly nuclear location, further enhancing this response [27,28]. Nevertheless, the anticipated induction of apoptosis was not observed. HPV16 E6-specific single-chain variable fragments (scFvs) (i.e., antibody variable heavy (VH) and light (VL) chain domains joined together by a synthetic linker peptide) which are amenable to ectopic expression inside mammalian cells as intrabodies as well as to passive nuclear diffusion have also been explored [29,30,31,32]. However, because non-specific effects in HPV-negative cells as well as unexpected downstream cellular responses were observed [31,32], further research into their application is needed.

While mAb-based therapeutic molecules have been clinically approved over the last few decades for various malignancies [33], similar translation for HPV16-related cancers has not occurred, owing to challenging biological aspects of the HPV16 E6 protein itself, including its primarily nuclear location and potentially limited target epitope availability, as well as to technical hurdles in producing optimally functional antibodies or fragments thereof. Instead, antibody fragments derived from unconventional sources, such as IgGs produced by *Camelidae* species (e.g., llamas) which were discovered to naturally lack both light chains and CH1 domains (heavy-chain-only antibodies: HCAbs) in a subset of their antibody repertoire [34], may present useful, unexplored options. The variable domains of these HCAbs (denoted as VHHs) can be isolated as single-domain antibodies, which conveniently retain the complete ability of the full-size antibody to interact with its antigen and demonstrate affinities for target antigens similar to those of conventional antibodies [35,36]. Due to their small size (~15 kDa) and the hydrophilic amino acid substitutions which evolved at the absent VL interface [37,38], VHHs possess several properties which may prove beneficial for E6-targeting including convex paratopes which can interact with antigen epitopes inaccessible to conventional mAbs and scFvs [36,37,39,40,41], robust thermal and chemical stability [42,43], as well as the ability to facilely enter the nucleus through nuclear pores [29]. With superior solubility and efficient folding compared to conventional mAb fragments, VHHs are particularly amenable to both high-yield periplasmic expression in *E. coli* followed by transfection of the purified molecules into HPV-infected cells as well as to direct intrabody expression [36,44,45]. Lastly, VHHs also share a greater sequence homology to human than murine VHs, minimizing the extent of humanization required for clinical translation [46,47].

New therapeutic applications for VHHs are continuously being identified [36,48], and several VHHs against targets involved in inflammatory/auto-immune, bone, neurological, hematological, oncological, and infectious diseases have already progressed to clinical trial evaluation [36]. In addition, a humanized VHH, Caplacizumab (Cablivi™), has recently been approved in Europe and in the USA for the treatment of acquired thrombotic thrombocytopenic purpura [49]. Accordingly, upon recognition of their utility in the context outlined above, we sought to add to this growing compendium by isolating VHHs against the HPV16 E6 protein. The scope of this paper was to describe the methodology of developing sdAbs rather than testing their potential effect on the endogenous E6 protein.

## 2. Results

### 2.1. A Specific Heavy-Chain IgG Response Was Induced Following Each Llama Immunization

To maximize our odds of isolating high-affinity VHHs targeting HPV16 E6, we performed two separate llama immunizations with different recombinant antigens. The first llama (Immunization #1) was injected with His6-GenScript E6: a variant of HPV16 E6 corresponding to that found in the cervical carcinoma-derived HPV16-positive cell line CaSki [50,51,52] with an N-terminal His6 tag (Figure 1C). The second llama (Immunization #2) was injected with a mixture of His6MBP-4C/4S E6 and His6MBP-F47R 4C/4S E6, which are solubility-enhanced mutants of HPV16 E6 each with an N-terminal His6-MBP tag followed by a tobacco etch virus (TEV) protease cleavage site [15,53,54,55] (Figure 1C). These immunization approaches were intended to be complementary, as the first used a variant of E6 which was naturally occurring but minimally soluble in recombinant form and had only a small His6 tag added; whereas the second used lab-engineered mutants of E6 which demonstrated improved recombinant solubility, but also had a large (MBP; ~42 kDa) fusion add-on as well as the His6 tag.

Induction of the desired antibody responses was initially assessed by screening the total llama serum by an enzyme-linked immunosorbent assay (ELISA). As anticipated, serum collected 35 and 42 days following the initial immunization contained increased amounts of IgG which reacted with the injected recombinant antigen(s) compared to pre-immune serum collected on Day 0, for both Immunization #1 (Figure 2A) and Immunization #2 (Figure 2B). The pre-immune serum from Immunization #2 also contained some IgG with minor reactivity for the His6MBP-E6 proteins. This may perhaps be the result of a previous immune encounter the animal had with the E6 protein of a distantly related llama papillomavirus [56,57] or, more likely, with the bacterial MBP [58].

Next, Day 0 and Day 42 total serum from both immunizations was fractionated using protein A and protein G affinity chromatography [59,60] (Figure 3A). For simplicity, Day 35 serum was omitted from this step as its total response was similar to that of Day 42 serum in both instances. Subsequent ELISAs confirmed that in addition to conventional IgG, heavy-chain IgG were also involved in the observed positive total serum responses for both Immunization #1 (Figure 3B) and Immunization #2 (Figure 3C). In particular, for Immunization #1, only the Day 42 heavy-chain IgG3 (G1) fraction displayed a notable increase in reactivity for the injected recombinant antigen compared to its corresponding Day 0 fraction. Whereas, for Immunization #2, an increase was observed in all three heavy-chain IgG3 (G1), IgG2a (A1), and IgG2b/c (A2) fractions. Different amounts of contaminating IgM in the Day 42 versus Day 0 A2 serum fractions, which co-elutes during the chromatography process, may have skewed our ability to detect a positive heavy-chain IgG2b/c response in the case of Immunization #1. However, generally weak heavy-chain IgG2a/b/c immune responses have also been observed by us previously and have not affected our ability to isolate high-affinity VHHs from the resulting phage display libraries.

### 2.2. VHH Phage Display Libraries Corresponding to Each Llama Immunization Were Successfully Constructed and Enriched for Antigen-Specific Binders Using Subtractive Panning

Following confirmation of heavy-chain IgG responses, a VHH phage display library corresponding to each llama immunization (i.e., Library #1: His6-GenScript E6 immunization and Library #2: His6MBP-E6 immunization) was constructed from frozen lymphocytes collected on Day 42. Using well-established techniques [59,60,61], VHH DNA was PCR amplified, subcloned into the pMED1 phagemid vector [62] and transformed into TG1 *E. coli*. It was estimated that Library #1 and Library #2 had functional sizes of ~3.4 × 10^8^ and ~1.7 × 10^7^ transformants, respectively (as calculated by total transformants multiplied by the % of colonies determined to contain VHH inserts).

As the successful usage of recombinant HPV16 E6 in molecular biology applications was notably facilitated by the solubility-enhanced mutants [63], we chose to exclude His6-GenScript E6 from use as a target antigen during the subsequent panning and VHH characterization assays. Accordingly, each library was then individually panned against a mixture of both His6MBP-E6 proteins. To select against MBP-specific VHHs, a subtractive panning technique [64] was employed whereby the input phage were first incubated in a well coated with MBP before being transferred to His6MBP-E6 or PBS coated wells. Two rounds of panning were completed for each library, with the amount of protein in the MBP-coated subtractive well increased and the amount of protein in the His6MBP-E6-coated well decreased during the second round, to maximize selective pressure for high affinity, E6-specific VHHs. The blocking reagent was also switched during the second round, to prevent unintentional enrichment for VHHs with affinity to it.

### 2.3. The Eluted VHH-Displaying Phage Were Further Characterized Using Phage ELISA

Following panning, the VHH phagemid inserts of 48 random colonies from the round 1 eluted phage titer plates and of 96 random colonies from the round 2 eluted phage titer plates for each library were arbitrarily numbered and sequenced. Clones beginning with “A” or “C” were isolated from Library #1 or Library #2 round 1 eluted phage titer plates, respectively, and clones beginning with “2A” or “2C” were isolated from Library #1 or Library #2 round 2 eluted phage titer plates, respectively. VHH-displaying phage were then amplified for more than 40 clones with different complementarity-determining region 3 (CDR3) sequences (the main region of the VHH involved in interaction with the antigen [36]), and these VHHs were further characterized for their ability to bind to the His6MBP-E6 proteins by phage ELISA [59,60].

Of those investigated, 26 clones which generated a strong signal in wells coated with either His6MBP-E6 protein but not in wells coated with MBP or PBS were considered to be potential E6 binders and were selected for soluble expression and purification (Appendix A). The occurrences of additional clones with the same CDR3 sequences as these VHHs are detailed in Appendix A. Nine of our 26 candidates had CDR3 sequences which were identified multiple times and three (A47, C11, and C38) had CDR3 sequences which were identified in both libraries. Despite the implementation of subtractive panning techniques, most clones identified from the Library #2, round 1 and round 2 eluted phage titer plates were MBP binders which generated signal in wells coated with MBP, in addition to wells coated with either His6MBP-E6 protein. One of these clones, C26, was also selected for use as an MBP-binding control VHH in the forthcoming assays.

### 2.4. VHHs Were Expressed and Characterized Using Soluble ELISA

The 26 potential E6-binding clones identified by phage ELISA as well as the one MBP-binding clone (C26) were next expressed as soluble VHHs with a C-terminal HA-His6 tag, following subcloning of each corresponding DNA insert into the pSJF2H vector [65]. VHH expression was directed to the *E. coli* periplasm using an OmpA leader [66], which allowed for proper disulfide bond formation [67] as well as ease of extraction through osmotic shock rather than total cell lysis [60]. The VHHs were purified from the resulting extract using immobilized metal-ion affinity chromatography (IMAC) [61] (Figure 4A). Five of the 26 potential E6-binding VHHs (clones A26, C06, C27, 2A55, and 2A90) did not express with our overnight approach. The expression of clone A24 was also quite low relative to the other VHHs. Hence, due to limited starting material, these six clones were excluded from further analyses.

To initially gauge the ability of the purified VHHs to detect the His6MBP-E6 proteins, we performed an ELISA using 100 μg/mL of each clone as the primary antibody (Figure 4B). Due to the presence of a His6 tag on our antigens as well as our VHHs, we first tried an anti-HA + HRP secondary antibody. This, however, resulted in very weak signal with high non-specific background. Instead, we then used an anti-His6 + HRP secondary antibody. The background signal originating from the N-terminal His6 tags of the recombinant His6MBP-E6 proteins was accounted for by subtracting the absorbance of His6MBP-E6, MBP, or buffer-coated control wells, which were incubated with PBS instead of VHH, from the corresponding wells incubated with both a VHH and secondary antibody. The clones demonstrated varying levels of binding to both His6MBP-E6 proteins but, overall, the signal was notably weaker than that obtained by phage ELISA. This may be caused by less amplification of the signal from bound VHH by the anti-His6 + HRP secondary antibody relative to that generated by the anti-M13 + HRP secondary antibody, which targets a coat protein present in thousands of copies per phage [60]. It may also, in part, be an artefact of our subtractive method of analysis in this instance. As expected, C26 bound to MBP, in addition to both His6MBP-E6 protein.

### 2.5. Characterization of VHHs Using Western Blotting under Denaturing and Native Conditions

To narrow down our current pool of E6-binding clones to those we first sought to analyze using surface plasmon resonance (SPR), each VHH was tested in both reducing SDS-PAGE and native PAGE Western blots. This allowed us to further characterize their interactions with the recombinant His6MBP-E6 proteins in two additional, widely used molecular biology assays and provided insight into the ability of each VHH to detect a linear or conformational epitope (as similarly described by Hussack et al. [59,61]). Both blot types were repeated at least twice per clone, using purified VHH as the primary antibody followed by application of an anti-HA tag + HRP secondary antibody. A sample of purified VHH was also run on each gel, to rule out inconsistent functionality of the secondary antibody as a cause for any negative results. As a positive control, the HPV16 E6-specific 6F4 mAb [23] was also tested in these same assays, together with an anti-mouse + HRP secondary antibody.

In their current monomeric format, none of the potential E6-binding VHHs yielded a detectable signal under denaturing conditions. However, under native conditions, clone 2A17 yielded reproducible bands in the lanes loaded with either His6MBP-E6 protein but not in the lane loaded with MBP, indicating that it interacts with a conformational epitope on the E6 portion of the antigen. Comparatively, we found the 6F4 mAb bound to both denatured and native His6MBP-E6 proteins, indicating that it interacts with a linear E6 epitope [59], as was originally reported by Giovane et al. [23] and Choulier et al. [68]. Clone C26 yielded reproducible bands in lanes loaded with MBP, in addition to lanes loaded with either His6MBP-E6 protein, confirming its status as an MBP-binding VHH. It too appears to interact with a linear epitope, as indicated by signal in both types of blots. As expected, none of the secondary antibodies bound to our recombinant antigens in the absence of primary antibody (Figure 4C).

### 2.6. Surface Plasmon Resonance Analyses Demonstrated Several VHHs Bound Recombinant E6 with Nanomolar Affinity

Next, we sought to determine with what affinity clone 2A17 bound to the recombinant E6 proteins. Based on the above described Western blot results, we also randomly included clones A01, A05, A09, A46 and, 2A12 as expected negative controls, the HPV16 E6-specific 6F4 mAb as a positive control, as well as clone C26 as an MBP-binding control. The initial assay setup involved amine coupling of both His6MBP-F47R 4C/4S E6 and His6MBP-4C/4S E6 proteins as well as a control His6MBP-intimin protein to the SPR sensor chip. SEC-purified VHHs (all were SEC-purified except A09, due to its low concentration and yield) or the 6F4 mAb (not SEC-purified) were then flowed over the chip to assess “Yes/No” binding. No notable binding to any of the recombinant proteins by VHHs A01, A05, A09, A46, 2A12, or 2A17 was detected (data not shown). However, the 6F4 mAb bound to both His6MBP-F47R 4C/4S E6 and His6MBP-4C/4S E6 proteins and the C26 VHH bound to all three recombinant proteins containing MBP as expected (Figure 5A).

To rule out the possibility that amine coupling was impacting the integrity of the E6 proteins, a second SPR assay was performed where the VHHs or 6F4 mAb were amine-coupled to the SPR sensor chip and His6MBP-F47R 4C/4S E6 or His6MBP-4C/4S E6 proteins flowed over the chip. Using this assay format, E6-specific binding was observed for VHHs A05, 2A12, and 2A17 (they bound to both His6MBP-E6 proteins but not the control His6MBP-intimin protein) (data not shown). Clones A01 and A09 did not bind to any of the recombinant proteins (data not shown). As expected, the 6F4 mAb bound to both His6MBP-F47R 4C/4S E6 and His6MBP-4C/4S E6 proteins and clone C26 bound to all three recombinant proteins (data not shown). We hypothesized that the harsh coupling conditions in the initial assay setup may have disrupted the conformational epitopes detected by these VHHs but did not affect the binding of the 6F4 mAb, which, in contrast, detects a linear epitope at the N-terminus of the E6 protein [23,68]. Using this second assay orientation, single-cycle kinetics data were then collected for the VHHs to determine affinities and kinetics. Confirming and elaborating upon the “Yes/No” binding results, we found clones A05, 2A12, and 2A17 bound both His6MBP-F47R 4C/4S E6 and His6MBP-4C/4S E6 proteins with nanomolar range affinities (Figure 5B,C) but did not bind the control His6MBP-intimin protein.

As we were able to detect VHHs A05 and 2A12 binding the recombinant His6MBP-F47R 4C/4S E6 and His6MBP-4C/4S E6 proteins using SPR (a label-free assay) but not with our above described Western blots under native conditions (which rely on the VHH HA tags), we first confirmed the HA tags were still intact on our stored stocks of purified VHHs by resolving samples of each clone by reducing SDS-PAGE, transferring them to polyvinylidene difluoride (PVDF) and then probing the membranes with an anti-HA + HRP antibody (Appendix A). Next, the recombinant His6MBP-F47R 4C/4S E6 and His6MBP-4C/4S E6 proteins as well as the recombinant MBP protein were spotted onto nitrocellulose membranes in dot blot format and increased VHH concentrations were tested. We were able to obtain a detectable signal when 2x the initial concentration of 2A12 or 4× the initial concentration of A05 was applied as the primary antibody (Appendix A). Since this relates to the different affinities we determined for each clone using SPR, it appears that the native PAGE Western blots have less sensitivity than the SPR assays to detect the interactions between our VHH candidates and recombinant proteins. We then blotted the remaining clones and interestingly, all tested positive using 5.4 μg/mL for A34, C38 and 2A03, and 10.8 μg/mL for the remaining VHHs (A09, A27, A37, A45, A46, A47, C11, C36, 2A04, 2A10, 2A15, 2A51, and 2A78). The A01 VHH was not tested in this experiment as no more soluble antibody were available (Appendix A).

## 3. Discussion

For the past decade, the therapeutic potential of VHHs has been explored for several tumor viruses in certain contexts: the hepatitis B virus (HBV), hepatitis C virus (HCV) and HPV. The main concept for these studies was to impede the viral life cycle and virion production to protect the infected host from further organ damage. For example, VHHs against the S domain of HBV envelope proteins (HBsAg) inhibited viral particle secretion in a mouse model [69] and VHHs against the HBV nucleocapsid core protein (HBcAg) disrupted its subcellular localization [70]. However, the oncogenic influence of these proteins following HBV integration remains poorly understood [71] and the potential benefit of these VHHs in such a scenario is yet to be determined. In addition, VHHs against the HCV RNA-dependent RNA polymerase (NS5B) [72], helicase (NS3 C-terminus) [73], serine protease (NS3/4A) [74], membrane web and replication complex formation protein (NS4B) [75], and envelope glycoprotein (E2) [76] have been shown to inhibit viral replication or cell-to-cell transmission in vitro. Thus far, for HPV16, only VHHs against the major capsid protein (L1) [77,78] and the oncoprotein E7 [79] have been reported. However, there remains an unmet need for the development of therapeutic VHHs targeting the E6 oncoprotein, which we sought to address with this study.

As target antigen properties such as solubility have been shown to influence the outcome of single-domain antibody library screens [80,81] and recombinant HPV16 E6 is naturally prone to aggregation [63], it was initially anticipated that both immunizing with and panning against the solubility-enhanced His6MBP-E6 proteins would provide the best odds of isolating VHHs against native E6 protein epitopes. However, 21 of the 26 VHHs indicated by phage ELISA to be potential E6 binders had CDR3 sequences which were from Library #1 (Appendix A), suggesting that immunizing with His6-GenScript E6 and panning the corresponding library against the solubility-enhanced His6MBP-E6 proteins was, instead, a more effective strategy. Although a subtractive panning technique was implemented, the majority of VHHs isolated from Library #2 were MBP binders. Perhaps the immune response to the His6MBP-E6 proteins was skewed more towards the larger MBP fusion partner (~42 kDa) than the E6 protein (~18 kDa). Therefore, de novo panning optimization will be required to facilitate the successful isolation of E6-binding VHHs from Library #2.

Of the 21 clones which were expressed as soluble VHHs and the subset further characterized with SPR, we have thus far identified that A05, 2A12, and 2A17 bind recombinant HPV16 E6 with affinities in the nanomolar range. Further investigation will then be needed to determine whether these VHHs will similarly bind the endogenous E6 protein (e.g., derived from HPV16-positive biological samples) using immunocytochemistry and a novel, quantitative dot blot assay [82]. We will then test all clones with SPR that test positive with the endogenous E6.

It will also be imperative to characterize what functional downstream effects they will elicit. One of the key protein-protein interactions which would be therapeutically beneficial to disrupt is that between E6 and the hijacked cellular E3A ubiquitin-protein ligase E6AP which, most notably, leads to viral-induced degradation of the tumor suppressor protein p53 [15]. Loss of p53 interferes with the ability of infected cells to respond to a range of stressors including DNA damage, hyperproliferation, hypoxia, and oxidation (as reviewed by Bieging et al. [83]) and is thought to prevent cell cycle modulation by the HPV16 E7 protein from inducing apoptosis [3]. Such an approach has been favored by others who have attempted to functionally inhibit the HPV16 E6 protein using scFvs [30,31,32]. Molecules derived from sources other than natural immune system components have additionally been investigated and include zinc-ejecting compounds [84,85], small compounds [86,87,88], inhibitory peptides [89,90,91,92,93], peptide aptamers [94], as well as bivalent inhibitors [55,95]. VHHs targeting the C-terminus of the E6 protein are also of therapeutic interest for their potential to disrupt interactions with various host PDZ-domain containing proteins such as those involved in cell polarity and cell–cell signaling (as reviewed by Klingelhutz and Roman [13]). The utility of this concept was similarly demonstrated in a study by Belyaeva et al. [96] using RNA aptamers. Hence, future work is needed to determine which E6 terminus is targeted by each of our VHH candidates and whether they can be used in combination to target both termini at once.

The previously reported E6-inhibitory molecules have had varying levels of success in decreasing cell proliferation and inducing apoptosis while maintaining minimal non-specific effects on HPV-negative cells and, notably, only one study reported in vivo data [32]. In addition, therapeutic responses did not always occur in the manner expected. For example, although the solubility-improved 1F4 scFv (adapted from the 1F1 and 6F4 mAbs [23]) was able to induce apoptosis when expressed as an intrabody in HPV16-positive cell cultures, it also non-specifically decreased proliferation in HPV-negative cells and the apoptotic pathway involved was not fully elucidated [31]. The recent intrabody study characterizing the NLS-conjugated 17nuc scFv, which was isolated by Intracellular Antibody Capture Technology, also reported a necrotic response both in vitro and in vivo, rather than apoptosis [32]. Interestingly, the E6/E6AP interaction-inhibiting compounds studied by Malecka et al. [88] did not reduce the proliferation of the HPV16-positive cell line SiHa, despite restoring p53 protein levels, but did in PA-1 cells transfected with HPV16 *E6*. Such discord between p53 restoration and the subsequent induction of apoptosis in metastatic cervical carcinoma-derived HPV16 cell lines has also been discussed in the context of E6 suppression by siRNA [21], leading us as well as others [88] to question whether decades of in vitro culture have introduced subsequent mutations in apoptotic pathways which reduce reliance on continuous oncogene expression. Until it can be reliably demonstrated that CRISPR/Cas9 disruption of *E6* at the DNA level [19] induces apoptosis in such cell lines without disruption to other host genes, it will be difficult to discern whether this is indeed the case and to what extent it may apply to freshly derived patient samples. Nevertheless, p53 restoration (especially when not mutated), even without inducing apoptosis, may still provide therapeutic benefits, due to the additional roles this protein plays in regulating immune response [97] and in causing cellular senescence [83]. Although senescent cells (i.e., cells which exhibit permanent growth arrest but remain viable) secrete proinflammatory molecules such as cytokines and their accumulation in large numbers may undesirably counteract tumor regression [98,99].

A recent study by Stevanović et al. [100] unexpectedly found that within populations of infused T cells which were cultured to enhance reactivity against the viral E6 and E7 proteins, it was tumor-infiltrating lymphocytes targeting neoantigens which played a key role in the regression of patients with metastatic HPV-positive cervical cancer. Although this study presents the results of only two patients (one HPV16-positive and one HPV18-positive) it indicates that added utility may be derived from simultaneously administering VHHs isolated against the viral E6 antigen together with those isolated against such patient-specific, non-viral targets. Their inclusion would also further increase the personalized aspect of our proposed therapeutic approach. With neoantigen expression vastly restricted to HPV-infected cells, such combinatorial treatment would still be anticipated to minimally affect surrounding healthy tissues. However, the timespan required to complete the VHH isolation and characterization process for each individual patient may currently be prohibitive, potentially restricting the applicability of this combinatorial treatment approach to more severe lesions.

With these concepts in mind, the further functional characterization of our VHHs in monolayer cell cultures, three-dimensional raft cultures, as well as small animal models (as similarly described by Togtema et al. [21] for siRNA) can be expected to identify their most optimal mode of application. The development of effective, clinically compatible strategies for the intracellular delivery of the soluble VHHs themselves will also be required. Various antibody delivery techniques have been summarized by others [36,45,101] and circumvent the patient risks associated with intrabody expression plasmids. In particular, several studies have demonstrated success in linking VHHs to the cell penetrating peptide, penetratin [72,73,74,75]. Our group has also explored the use of a high intensity focused ultrasound (HIFU)-based sonoporation technique to deliver HPV16 E6 mAbs [20]. It works by creating temporary pores in the cell membrane through HIFU-induced microbubble cavitation and can localize the delivery of therapeutic molecules to specific regions of target tissue [102]. Ideally, these approaches will need to be compared to determine which is most suitable in this context.

In addition to therapeutic limitations, researchers have also encountered difficulties using commercial mAbs for the immunodetection of HPV16 E6 and instead often examine downstream protein levels (e.g., p53) as proxies [103,104,105,106,107]. Immunocytochemistry has been particularly challenging [92,108], with limited success requiring the application of very specific staining protocols and complex image thresholding techniques [108]. However, as reviewed by Beghein and Gettemans [109], the stability and antigen accessibility of VHHs have already facilitated their use for the immunofluorescent detection of various cellular antigens including vimentin [110], β-catenin [111], histone H2A-H2B heterodimer [112], Vγ9Vδ2 T-cell receptor [113], and leukocyte receptor ChemR23 [114]. It remains to be seen if our VHH candidates likewise demonstrate utility in these diagnostic types of applications. Thus, we ultimately envision the isolation of HPV16-E6 specific VHHs as a key first step, with the full range of their potential applications compared to that of traditional mAbs and fragments thereof to hopefully be fully realized in subsequent investigations.

## 4. Materials and Methods

### 4.1. Recombinant Proteins

Three types of recombinant HPV16 E6 protein were used in this study (Figure 1C). The first (denoted His6-GenScript E6) corresponded to the protein variant found in the cervical carcinoma-derived HPV16-positive cell line CaSki, which contains R10G and L83V amino acid substitutions from the reference sequence (GenBank Accession #: K02718.1) [50,51,52]. It was expressed with an N-terminal polyhistidine (His6) tag in *E. coli*, solubilized from inclusion bodies, purified using IMAC, and dialyzed into buffer containing 1× PBS, pH 7.4, 1% sodium lauroyl sarcosine by GenScript (Piscataway, NJ, USA).

The other two recombinant HPV16 E6 proteins (denoted His6MBP-4C/4S E6 and His6MBP-F47R 4C/4S E6) contained the solubility-enhancing amino acid substitutions C80S, C97S, C111S, C140S, or F47R, C80S, C97S, C111S, C140S, respectively, and were expressed by us from the previously described pETM-41 plasmids (kindly provided by Dr Gilles Travé, IGBMC, France) with an N-terminal His6-maltose binding protein (MBP) tag followed by a TEV protease cleavage site [15,53,54,55]. Briefly, expression of each His6MBP-E6 protein was induced with 1 mM Isopropyl β-d-1-thiogalactopyranoside (IPTG) overnight at 15 °C in ClearColi^®^ BL21 (DE3) *E. coli* (Lucigen; Middleton, WI, USA; Cat. #: 60810-1) grown in Luria-Bertani (LB) media. The bacterial pellets were resuspended in ice-cold lysis buffer (50 mM Tris-HCl pH 8.0, 25 mM NaCl) and stored for 1 h, at −80 °C or overnight at −20 °C. SigmaFAST™ EDTA-Free Protease Inhibitor Cocktail Tablets (Sigma-Aldrich; Oakville, ON, Canada; Cat. #: S8830) and 2 mM dithiothreitol (DTT) were then added, and the thawed cells lysed using 150 μg/mL lysozyme. The lysate was further incubated with 60 units/mL DNase I and then clarified by centrifugation. Finally, the recombinant antigen was purified from the clarified lysate using IMAC and dialyzed into buffer containing 50 mM Tris-HCl pH 6.8, 400 mM NaCl, 2 mM DTT.

Recombinant MBP in 20 mM Tris pH 8.0 was purchased from Abnova (Walnut, CA, USA; Cat. #: P4989). All proteins were kept at −80 °C for long-term storage.

### 4.2. VHH Phage Display Library Construction, Selection, and Screening

For llama immunizations, serum fractionizations at Day 0 and 42, construction of VHH phage display libraries, subtractive panning, and phage ELISA, see Appendix A.

### 4.3. VHH Expression and Purification

DNA inserts consisting of an OmpA leader (to direct VHH expression to the periplasm of *E. coli* [66]) + VHH + human influenza hemagglutinin (HA)-His6 tag coding sequences were synthesized for 26 VHHs which yielded notable E6-specific phage ELISA signals as well as for one VHH which demonstrated MBP-binding, and these inserts were subcloned into the pSJF2H expression vector [65] via the *Eco*RI and *Bam*HI restriction sites through services available at GenScript. Aliquots of 5 μL Zymo Mix & Go!™ chemically competent TG1 *E. coli* (Cedarlane Laboratories; Burlington, ON, Canada; Cat. #: T3017) were transformed with ~50 ng insert-containing pSJF2H vector, according to supplier instructions. The transformed cells were grown in 250 mL 2xYT media supplemented with 100 μg/mL ampicillin and 0.1% (*w*/*v*) glucose and VHH expression induced with 0.8 mM IPTG overnight at 37 °C. Periplasmic extraction using TES buffer, IMAC purification, and calculation of VHH concentration were completed as similarly described by Hussack et al. [61], except both incubations on ice during TES extraction increased from 30 min to 1 h. The purified VHHs were then aliquoted and stored in PBS at −20 °C long-term, following their analysis with reducing SDS-PAGE.

### 4.4. ELISA and Dot/Western Blot of Purified VHHs

To characterize the ability of the purified VHHs to detect recombinant HPV16 E6, ELISA, reducing SDS-PAGE and native PAGE Western blots, as well as dot blots were performed. For the ELISA, 0.5 μg of His6MBP-4C/4S E6, His6MBP-F47R 4C/4S E6, or MBP (100 μL/well; His6MBP-E6 antigens diluted in their storage buffer, MBP diluted in PBS) or 100 μL/well His6MBP-E6 protein storage buffer alone was coated in Nunc™ MaxiSorp™ 96-well plates overnight at 4 °C. The wells were blocked with 5% milk powder in PBS-T (PBS + 0.05% (*v*/*v*) Tween 20) at 37 °C, prior to application of 100 μg/mL of each VHH diluted in PBS (100 μL/well) for 1 h at room temperature. The wells were then washed with PBS-T and incubated with a rabbit anti-His6 + HRP antibody (Cedarlane Laboratories; Cat. #: A-190-114P) diluted 1:20,000 in PBS (100 μL/well) for 1 h at room temperature. The wells were again washed with PBS-T and then incubated with TMB (Mandel Scientific; Guelph, ON, Canada; Cat. #: KP-50-76-00). Briefly, TMB substrate was added to the wells for approximately 5 min. The reaction was stopped by the addition of 100 μL/well 1 M phosphoric acid and the absorbance read at 450 nm.

For Western blots run under denaturing conditions, 10 ng/lane of His6MBP-4C/4S E6, His6MBP-F47R 4C/4S E6, or MBP as well as 5 ng/lane HAHis6-tagged VHH (as a positive secondary antibody control) were separated on Mini-PROTEAN^®^ TGX™ precast 4–20% gradient polyacrylamide gels (BioRad; Mississauga, ON, Canada; Cat. #: 4561096) for ~70 min at 120 V and transferred to PVDF membrane for 1 h at 100 V. The membranes were blocked with 5% milk powder in TBS-T for 1 h at room temperature followed by overnight 4 °C incubation with 2.7 μg/mL purified VHH or mouse monoclonal anti-HPV16 E6 antibody (clone 6F4; kind gift from Arbor Vita Corporation; Fremont, CA, USA) diluted in blocking buffer. As this concentration of mAb yields reproducible detection from HPV16-positive cell lysates in our lab [21], it was employed as our initial test concentration. The membranes were washed with TBS-T, followed by incubation with a goat polyclonal anti-HA tag + HRP secondary antibody (Abcam; Toronto, ON, Canada; Cat. #: ab1265) diluted 1:5000, a mouse monoclonal anti-HA tag + HRP secondary antibody (Thermo Fisher Scientific; Cat. #: 26183-HRP) diluted 1:1000–1:2000 (depending on individual lot characteristics), or a goat anti-mouse IgG + HRP secondary antibody diluted 1:2000 in blocking buffer for 1 h at room temperature, as appropriate. The membranes were again washed in TBS-T and chemiluminescence detection was completed as described in [21]. All VHHs were tested under these conditions in a minimum of two separate experiments.

For the native PAGE Western blots, 5 μg/lane of His6MBP-4C/4S E6, His6MBP-F47R 4C/4S E6, or MBP as well as 2.5 μg/lane HA-His6-tagged VHH (as a positive secondary antibody control) were separated on 10% polyacrylamide gels without SDS for 2.5 h at 100 V, 4 °C and transferred (in buffer without methanol) to PVDF membrane for ~16 h at 20 V, 4 °C. Immunoblotting was then performed as described above. All VHHs were tested under these conditions in two separate experiments.

For the dot blots, 2 µg of His6MBP-4C/4S E6, His6MBP-F47R 4C/4S E6, or MBP was spotted on a nitrocellulose membrane and allowed to dry for 1 h at room temperature. The membrane was then blocked with 5% milk powder in PBS-T (0.01% (*v*/*v*) Tween 20) for 1 h at room temperature followed by overnight 4 °C incubation with 2.7–10.8 μg/mL purified VHH diluted in PBS-T. The membranes were washed with PBS-T, followed by incubation with an anti-HA tag + HRP secondary antibody as described above. The membranes were again washed in PBS-T and chemiluminescence detection completed as indicated above.

### 4.5. Size Exclusion Chromatography and Surface Plasmon Resonance Analyses

Prior to SPR experiments, the VHHs and His6MBP-E6 proteins were size exclusion chromatography (SEC)-purified. The VHHs (300–500 µg) were purified using a Superdex 75 Increase 10/300 GL column (GE Healthcare; Mississauga, ON, Canada) while the His6MBP-E6 proteins (300 µg) were purified using a Superdex 200 Increase 10/300 GL column (GE Healthcare). A flow rate of 0.5 mL/min and HBS-EP+ buffer (10 mM HEPES pH 7.4, 150 mM NaCl, 3 mM EDTA, 0.05% (*v*/*v*) P20 surfactant) were used for all experiments. Fractions of 0.5 mL volume were collected, and the concentration determined through absorbance measurements at 280 nm.

All SPR analyses were performed on a Biacore T200 instrument (GE Healthcare) at 25 °C, using HBS-EP+ as running buffer and Series S CM5 sensor chips (GE Healthcare). All data were reference-flow-cell subtracted and analyzed using Biacore T200 software v3.0 (GE Healthcare). Two SPR assay formats were performed. In the first assay, E6 proteins (His6MBP-4C/4S E6, His6MBP-F47R 4C/4S E6) and a control MBP protein (His6MBP-intimin) were coupled to a CM5 chip through standard amine coupling. Briefly, ~1200 resonance units (RUs) of each protein were immobilized in 10 mM acetate buffer pH 5.0 (His6MBP-E6 proteins) or pH 4.5 (His6MBP-intimin), producing surfaces with theoretical *R*_max_s ranging from 260–300 RUs for the VHHs. VHHs (A01, A05, A09, A46, 2A12, 2A17, C26) were injected at a single concentration of 500 nM and the HPV16 E6-specific 6F4 mAb was injected at 100 nM, all at a flow rate of 50 µL/min for 180 s followed by 600 s of dissociation. Surfaces were regenerated with a 120 s pulse of 10 mM glycine pH 1.5 at a flow rate of 30 µL/min. In the second assay, VHHs (A01, A05, A09, 2A12, 2A17, C26) and the 6F4 mAb were coupled to Series S CM5 sensor chips using similar conditions reported above (10 mM acetate buffer pH 5.0). Approximately 400 RUs of each VHH or mAb were immobilized. Non-SEC-purified His6MBP-E6 or His6MBP-intimin was injected at a single concentration of 500 nM (over VHH surfaces) or 100 nM (over the mAb surface) to determine surface activity. Next, using single-cycle kinetics, a concentration range (500–31.3 nM) of SEC-purified His6MBP-4C/4S E6 or His6MBP-F47R 4C/4S E6 was injected over each surface at a flow rate of 40 µL/min for 180 s followed by 600 s of dissociation to determine affinities and kinetics. Sensorgrams were reference subtracted and fit to a 1:1 binding model. Surface regeneration conditions were identical to the first assay described above.

## Figures and Tables

**Figure 1 ijms-20-02088-f001:**
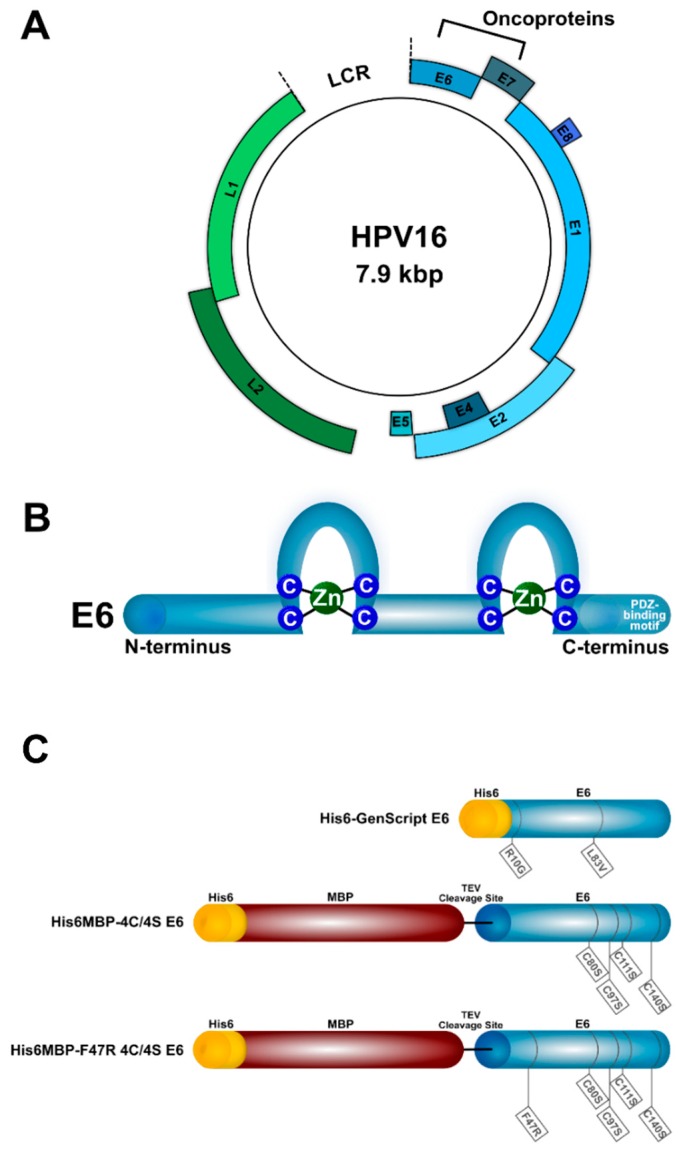
The HPV16 genome and E6 oncoprotein. (**A**) The circular, double-stranded DNA genome (~7.9 kbp) consists of a long control region (LCR) which regulates viral transcription, as well as seven early (*E1*, *E2*, *E4*, *E5*, *E6*, *E7*, and *E8*) and two late (*L1* and *L2*) genes. The proteins encoded by the early genes promote viral persistence, replication, and release, whereas those encoded by the late genes create the viral capsid [12,16,17]. The HPV16 genome (GenBank Accession #: K02718.1) was visualized using the Pathogen–Host Analysis Tool [18]. (**B**) The E6 oncoprotein (~18 kDa) contains two zinc-binding domains. It interacts with host proteins after first complexing with the hijacked ubiquitin ligase E6AP or via its C-terminal PDZ-binding motif, driving cancerous changes in infected cells [13,14]. (**C**) Schematic representation of the recombinant HPV16 E6 proteins used in this study. Their amino acid differences from the reference sequence (GenBank Accession #: K02718.1) as well as their respective polyhistidine (His6) tags and maltose binding protein (MBP) fusion partners are indicated.

**Figure 2 ijms-20-02088-f002:**
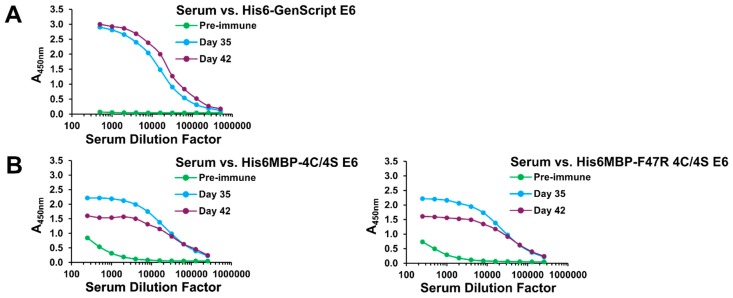
Total serum response following each llama immunization. Following both Immunization #1 (**A**) and Immunization #2 (**B**), ELISAs demonstrated that serum collected on Days 35 and 42 contained an increased amount of IgG which reacted with the injected recombinant antigen(s) compared to pre-immune serum collected on Day 0, indicating specific immune responses had been induced.

**Figure 3 ijms-20-02088-f003:**
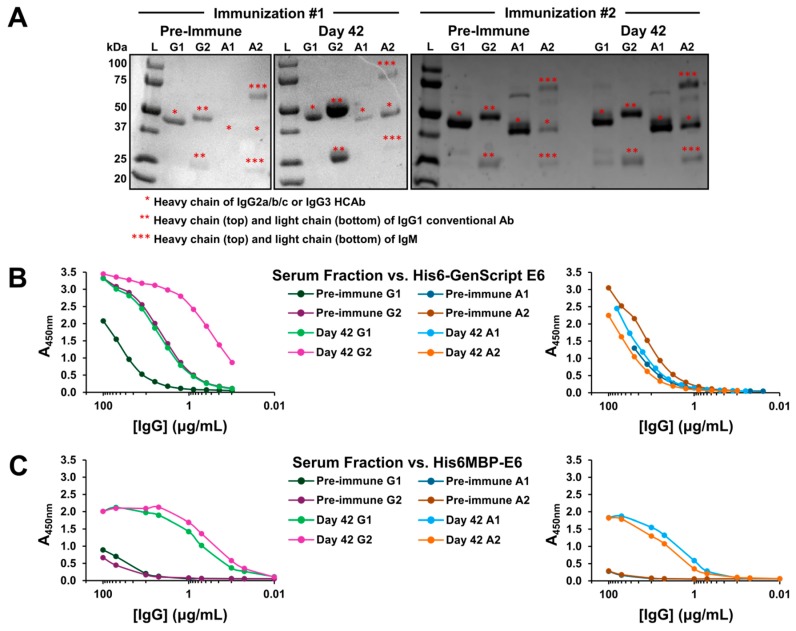
Serum fractionation and confirmation of a heavy-chain IgG (HCAb) immune response following each llama immunization. (**A**) Reducing sodium dodecyl sulfate-polyacrylamide gel electrophoresis (SDS-PAGE) was used to analyze the purity of the pre-immune and Day 42 serum fractions: G1 fraction = IgG3 HCAb, G2 fraction = IgG1 conventional antibody, A1 fraction = IgG2a HCAb, and A2 fraction = IgG2b/c HCAb. Approximately 2.0-2.5 μg of sample was loaded in each lane, except for some of the more dilute A1 and A2 fractions for which smaller, but still visualizable, amounts were loaded. IgM which sometimes co-elutes with IgG2b/c was noted in all A2 fractions examined here. In the Immunization #2 pre-immune and Day 42 A1 serum fractions, an additional band is visible below the 75 kDa marker which may consist of residual unreduced HCAb. (**B**) ELISAs demonstrated that Immunization #1 Day 42 G1 and G2 serum fractions showed increased recognition for the injected recombinant antigen compared to the corresponding pre-immune serum fractions. (**C**) An increased response was also observed for all four Immunization #2 Day 42 serum fractions.

**Figure 4 ijms-20-02088-f004:**
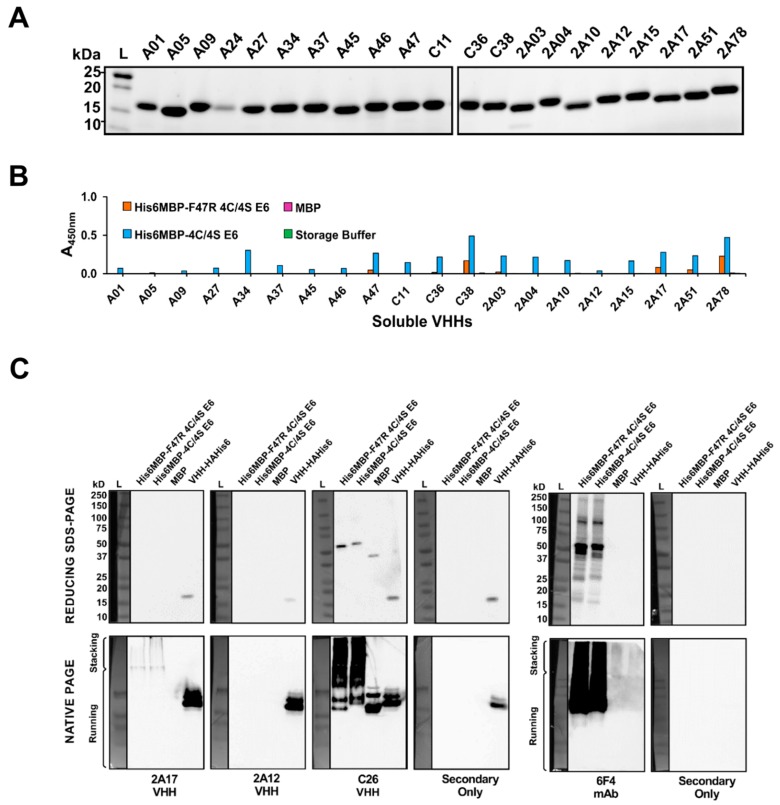
VHH purification and determination of their ability to detect recombinant HPV16 E6. (**A**) Reducing SDS-PAGE was used to analyze the purity of the VHHs following IMAC purification. Approximately 3 μg of sample was loaded in each lane, except for the more dilute clones A01, A24, C38, and 2A17 for which amounts ranging from approximately 1–2.5 μg were loaded. (**B**) An ELISA using 100 μg/mL of each clone demonstrated that the purified VHHs were functional but varied in their ability to detect the His6MBP-E6 proteins. (**C**) Reducing SDS-PAGE and native PAGE Western blots. When considered altogether, the assay results indicated clone 2A17 interacts with a conformational epitope on the E6 portion of the His6MBP-E6 proteins; whereas none of the other purified VHHs yielded detectable signal (clone 2A12 shown as an example). In contrast, the HPV16 E6-specific 6F4 mouse mAb interacts with a linear E6 epitope. Clone C26 was also further confirmed as an MBP binder which likely also interacts with a linear epitope. For reducing SDS-PAGE, 10 ng/lane of His6MBP-F47R 4C/4S E6, His6MBP-4C/4S E6, and MBP as well as 5 ng/lane of purified VHH 2A04 was loaded. For native PAGE, 5 μg/lane of His6MBP-F47R 4C/4S E6, His6MBP-4C/4S E6, and MBP as well as 2.5 μg/lane of purified VHH A46 was loaded. Note, for the native PAGE Western blots, a ladder was used only as an indicator of protein transfer and does not reflect the molecular weight of the resolved proteins.

**Figure 5 ijms-20-02088-f005:**
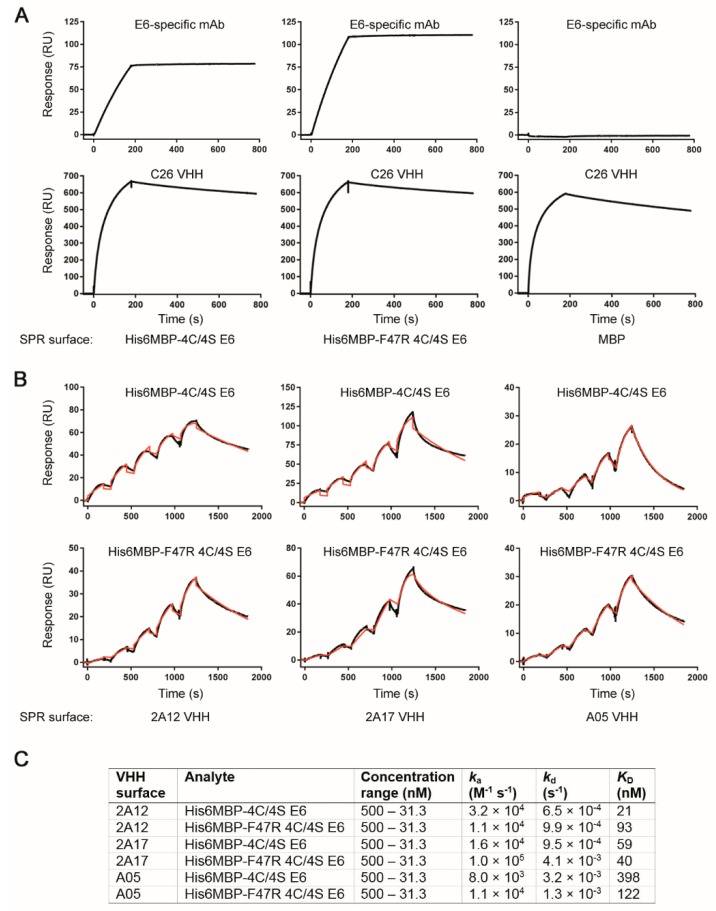
Surface plasmon resonance (SPR) binding assays demonstrating the interaction of the His6MBP-F47R 4C/4S E6 and His6MBP-4C/4S E6 proteins and VHHs. (**A**) SPR surfaces with His6MBP-F47R 4C/4S E6, His6MBP-4C/4S E6 or control MBP (His6MBP-intimin) were created and VHHs or mAbs flowed over. The HPV16 E6-specific 6F4 mAb bound E6-containing surfaces and not the MBP control surface while the C26 VHH bound all MBP-containing surfaces. All other VHHs tested did not bind any surface (data not shown). (**B**) VHHs were immobilized and SEC-purified His6MBP-F47R 4C/4S E6 or His6MBP-4C/4S E6 proteins flowed over using single-cycle kinetics to determine affinities and kinetics. Black lines represent raw data and red lines represent 1:1 binding model fits. (**C**) Summary of VHH binding affinities calculated from the sensorgrams in (**B**).

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
