# Peer review of "Single-Domain Antibodies Represent Novel Alternatives to Monoclonal Antibodies as Targeting Agents against the Human Papillomavirus 16 E6 Protein"

_ijms, 2019, doi:10.3390/ijms20092088_

Reviewer 1 Report

The authors present a solid strategy to construct and test single domain antibodies (VHH) to target HPV16 E6 oncoprotein. HPVs account for over >5% of human malignancies and the E6 oncoprotein has been documented to play roles not only in the onset and progression of carcinogenesis but also in its maintenance. Thus, the development of effective targeting agents is of profound significance with obvious potential for clinical application.

The authors have used a rational strategy to identify E6 -targeting VHH by immunizing llamas with different recombinant E6 proteins, validating successful targeting of E6 recombinants by the sera, and constructing VHH phage display libraries using frozen lymphocytes. This allowed the identification and purification of VHH from various clones, 21 of which were further tested for their ability to bind recombinant E6. A subset of those were tested in SPR assays.

The characterized clones did not have consistent affinity for recombinant E6 in all assays used, but the reasons for this are only briefly addressed in paragraph 2 of the discussion. In particular, the rationale of the authors testing only 3 particular clones using SPR seems unclear, since the previous assays do not seem to clearly point to those specific clones. I would suggest that more clones would need to be tested.

I agree with the authors that the ability to detect native E6 is a feature that maybe useful for research and diagnostic applications in a much shorter time-frame than expected for potential clinical applications. With that in mind I would suggest that the authors present some data regarding the ability of some of the identified clones to bind endogenous E6.

Overall, I find that the work described is sound and the authors present their data in an organized and well-written manuscript. Nevertheless, I suggest that they need to present data on the binding affinities for more clones, as well as some data on the binding affinity for endogenous E6.  This would validate the future potential of their strategy.

Author Response

Please see the file in the attachment.

Reviewer 2 Report

The paper describes the selection and characterization of single-domain antibodies against the HPV16 E6 oncoprotein and suggests that these may represent a therapeutic alternative to the more "classic" monoclonal antibodies to counteract the tumor activity of the E6 oncoprotein.

The manuscript is well detailed and well written, and the characterization is done according to the universal protocols as it should be.

Nevertheless, some criticisms are evident:

-The reason why the unmet need for a therapy for HPV lesions is to be solved by targeting just the E6 and just using single-domain antibodies is not clear, and the authors should better explain it.

-Single-domain antibodies are certainly an interesting possibility to cover, although those whose selection is reported in the paper seem not to work adequately in recognizing the E6 antigen in the immunological assays performed.

-A proliferation test on HPV-positive cells expressing the anti-E6 single-domains antibodies would not be too laborious and might demonstrate the hypothesis.

-In pursuing extreme precision in the description of experimental work, there are probably repetitions in results and methods, which should be revised and merged if possible.

-In general, the paper is quite long, and shortening it slightly may increase its readability.

Specific points:

Introduction section.

Page 2, lines 88, 89: it is not clear what the authors refer to when they mention unwanted side effects and unexpected downstream cellular responses of other anti-E6 antibodies described in the literature. If they refer to the toxic effect of scFv aggregation due to poor solubility (ref 31), this is not a general criticality but only means that solubility is an important feature for the functioning of these molecules.

Results section.

Paragraph 2.4, Fig. 4B and page 7, lines 239-247: the ELISA performed to ascertain the single-domain antibody capacity of detecting the recombinant E6s was performed at a very high concentration of primary antibody (100 mg/mL); despite this, the signals obtained with anti-His6 with a rather artificial method, namely after subtraction of the signal ascribed to His-E6s recognition were, as admitted by the authors themselves, very weak. The situation is not improved by the reason given, because it is quite obvious that these signals are weaker than those obtained with phage ELISA, since phages act as amplifiers.

Paragraph 2.5, Fig. 4C and lines 281-283: in characterization of VHHs by WB, it is claimed that 2A17 yields reproducible bands corresponding to the HisMBP-E6s, meaningful of interaction with conformational epitopes of the proteins. Nevertheless, only faint bands at the limit between stacking and running gel are visible, raising doubts on the actual VHHs capacity to recognize their antigen. However, to dispel any doubt, it would be useful to know if the expected anti-proliferative effect is actually obtained using such VHHs, at least in vitro.

Discussion section.

Lines 449-450: the VHHs selected do not seem to be very reactive towards the E6, thus it is quite questionable that they could find application in diagnostics.

Author Response

Please see the file in the attachment.

Round  2

Reviewer 1 Report

In light of the corrections made by the authors I find that the manuscript in now fit for publication.